# FairLoRA: Unpacking Bias Mitigation in Vision Models with Fairness-Driven Low-Rank Adaptation

## Abstract

Recent advances in parameter-efficient fine-tuning methods, such as Low Rank Adaptation (LoRA), have gained significant attention for their ability to efficiently adapt large foundational models to various downstream tasks. These methods are appreciated for achieving performance comparable to full fine-tuning on aggregate-level metrics, while significantly reducing computational costs. To systematically address fairness in LLMs previous studies fine-tune on fairness specific data using a larger LoRA rank than typically used. In this paper, we introduce FairLoRA, a novel fairness-specific regularizer for LoRA aimed at reducing performance disparities across data subgroups by minimizing per-class variance in loss. To the best of our knowledge, we are the first to introduce a fairness based finetuning through LoRA. Our results demonstrate that the need for higher ranks to mitigate bias is not universal; it depends on factors such as the pre-trained model, dataset, and task. More importantly, we systematically evaluate FairLoRA across various vision models, including ViT, DiNO, and CLIP, in scenarios involving distribution shifts. We further emphasize the necessity of using multiple fairness metrics to obtain a holistic assessment of fairness, rather than relying solely on the metric optimized during training.

## 1 Introduction

The advent of foundational models Bommasani et al. (2021) has led to the widespread adoption of parameter-efficient fine-tuning (PEFT) methods such as Low-Rank Adaptation (LoRA) Hu et al. (2021), enabling efficient adaptation to various downstream tasks. These methods offer significant computational advantages and often achieve performance comparable to full fine-tuning (FFT) of the entire model on aggregate metrics Hu et al. (2021); Zhao et al. (2024); Dettmers et al. (2024). However, their impact on fairness remains under-explored. More importantly, given the widespread use of LoRA for fine-tuning foundational models, the uncertainty regarding the impacts on fairness as well as bias mitigation, complicates deployment and raises ethical concerns, emphasizing the need to measure and mitigate disparate impacts.

Recent works on Fairness with PEFT has focused on either (i) finding the right parameters to tune through heuristic search algorithms Dutt et al. (2023) or (ii) fine-tuning on specific datasets curated for fairness Das et al. (2024). As noted in the papers, finding the right set of parameters is a challenging and often a computationally expensive problem, thereby contradicting the major advantage of using PEFT. Although effective, fine-tuning with fairness specific datasets can also be challenging given the complexities involved in data collection, curating, and labelling. Furthermore, it is also important to note that in order to mitigate bias based on different notions, we might need completely different datasets– thereby making it hard to scale.

The fairness community in ML has often argued the importance of the 'right metric' to measure unfairness and how that changes the answer to the question "Does X have disparate impact" Hashemizadeh et al. (2023); Simson et al. (2024). Our work emphasizes the importance of evaluating multiple fairness metrics rather than relying on a single measure. By considering metrics such as aggregate accuracy, minimum and median F1 scores across groups, and performance disparities between groups, we aim to capture a holistic view of both performance and fairness. This comprehensive

evaluation allows us to assess whether PEFT methods like LoRA consistently meet fairness standards or may lead to adverse outcomes in certain configurations.

We introduce **FairLoRA**, a novel fairness based LoRA that scales across models and datasets. Our findings indicate that FairLoRA performs comparable to or better than using fairness specific regularization with FFT, across most metrics. Our method is an in-processing bias mitigation method that aims at altering the learning objective with almost zero addition in computational cost, as opposed to heuristic search based fair finetuning.

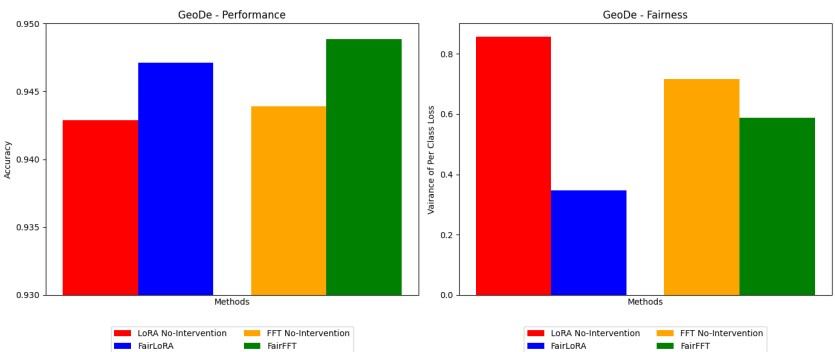

Figure 1: We compare the CLIP models with and without fairness regularizers for both full fine-tuning (FFT) as well as LoRA. *Dataset: GeoDE*. On the left. we notice that **FairLoRA has better overall performance** compared to LoRA. On the right, we visualize the effect on the variance of loss across classes and **FairLoRA has a lower variance** compared to all other methods. More detailed results and comparisons can be found in section 6.2.

Additionally, we explore the hypothesis that distribution shifts between pre-training and fine-tuning datasets contribute to fairness disparities. By analyzing models such as CLIP Radford et al. (2021), DINO Caron et al. (2021), and ViT Dosovitskiy (2020), we assess how pre-training strategies and data distributions affect fairness during fine-tuning. To investigate this, we conduct experiments on diverse datasets—Aircrafts Maji et al. (2013), GeoDE Ramaswamy et al. (2024), and Water-birds Sagawa et al. (2019)—which differ significantly from popular pre-training datasets. Our experiments show that distribution shifts can exacerbate fairness issues, but FairLoRA is able to successfully mitigate them.

Through this work, we aim to determine which approach—LoRA or Full Finetuning (FFT) —is fairer under different conditions and how fairness regularization affects performance. Our findings suggest that FairLoRA is superior to LoRA, especially when considering across metrics. It is also important to note that FairLoRA is also comparable to Fair FFT.

Our main contributions are as follows:

- We highlight the importance of comprehensive evaluation across multiple metrics when assessing fairness.
- We are the first to formulate - FairLoRA - a fairness regularizer based on reducing the variance among intra-class losses to improve the fairness of models fine-tuned with LoRA.
- We demonstrate that higher ranks are not necessarily required to improve fairness through learning with FairLoRA.
- Our experiments show that FairLoRA can reliably improve fairness with across multiple architectures, with datasets that have distribution shift, and across LoRA ranks.

## 2  RELATED WORKS

**PEFT:** LoRA stands out among PEFT methods due to its efficiency and simplicity in fine-tuning large models. Unlike adapters that add new layers, LoRA injects low-rank matrices directly into the weight updates of pre-trained models, keeping the number of parameters the same as the original

model (for inference). This makes LoRA easy to implement with significantly lower memory and computational costs while training, and constant deployment cost compared to the original model, all while maintaining performance comparable to full fine-tuning (FFT). Our work measures the disparate impact of PEFT in vision models and proposes a novel way to mitigate bias.

**LoRA and Fairness:** Recent studies on Low-Rank Adaptation (LoRA) highlight key trade-offs between fairness, safety and performance Das et al. (2024); Ding et al. (2024). Although the fairness impact of LoRA depends heavily on the base model and rank (Ding et al., 2024) did not notice any systemic disparate impact. In terms of bias mitigation, as the rank increases, LoRA's performance and fairness become comparable to full fine-tuning (FFT) Das et al. (2024). Similarly, (Dutt et al., 2023) proposed a fairness aware PEFT by heuristically searching for the right set of parameters to update. Our work is different from the above mentioned because (i) we analyze the disparate impact across datasets with distribution shifts, a challenging problem for LoRA Lermen et al. (2023), (ii) we focus the bias mitigation on vision and Vision language models while the previous work Das et al. (2024) focuses on LLMs and finally, (iii) our work focuses on changing the LoRA objective function make it more generic compared to fine-tuning with a fairness specific dataset or performing heuristic search to find the 'correct' tunable parameters.

**Fairness in vision models:** Independent of PEFT, fairness in machine learning models is a well studied problem (Dwork et al., 2012; Dieterich et al., 2016; Verma & Rubin, 2018; Mehrabi et al., 2021; Zemel et al., 2013; Zhao & Gordon, 2022). Enforcing fairness has mainly focused on imposing requirements such as demographic parity, equalized odds, equal opportunity (Hardt et al., 2016), accuracy parity (Agarwal et al., 2018; Berk et al., 2021), or combinations of these properties (Zafar et al., 2017; Lowy et al., 2021; Bakker et al., 2020; Shui et al., 2022) through ine-tuning, penalized objective or constrains. Our fairness regularizer is inspired from the formulations used by Tran et al. (2022); Hashemizadeh et al. (2023) and aim to reduce the per-class variance in the loss.

**Model Flexibility and Generalization:** Recent work (Shwartz-Ziv et al., 2024) shows that neural network architectures vary in how they fit data, potentially impacting fairness across demographic groups. Architectures like CNNs and ViTs exhibit different efficiencies when adapting to new tasks, indicating that architecture plays a crucial role in a models (fairness) outcomes. Our analysis focuses on understanding similar trends with respect to fairness for LoRA based fine-tuning.

## 3 PRELIMINARIES: LoRA

Low-Rank Adaptation (LoRA) Hu et al. (2021) is a parameter-efficient fine-tuning (PEFT) method that injects trainable low-rank matrices into the weight updates of a pre-trained model, significantly reducing the number of trainable parameters without compromising performance. The core idea of LoRA is to decompose the weight updates into two low-rank matrices, which are then trained to capture task-specific knowledge.

Consider a pre-trained subset of parameters $\theta_0 \in \Theta$, where $\Theta$ represents the full parameter space of the model. In LoRA, instead of updating $\theta_0$ directly, the weight update $\Delta\theta$ is parameterized as a product of two low-rank matrices $A \in \mathbb{R}^{d \times r}$ and $B \in \mathbb{R}^{r \times k}$, where $r \ll \min(d, k)$. Thus, the updated parameter matrix becomes:

$$\theta = \theta_0 + \Delta\theta = \theta_0 + AB$$

Here, $A$ and $B$ are the trainable matrices, and $r$ is the rank, controlling the parameter reduction. By making $r$ much smaller than $d$ and $k$, the total number of trainable parameters is reduced from $d \times k$ to $r \times (d + k)$, leading to substantial memory savings.

## 4 FAIRLoRA: FAIRNESS AWARE LoRA

**Does LoRA have a systemic disparate impact?** Our experiments, similar to the work from Ding et al. (2024), notes that there is no systemic disparate impact when doing LoRA compared to full finetuning. That is even if LoRA tends to under-perform for a certain set of datasets, model, and metric combination, there is no specific pattern of disparate impact. These trends also hold for the lowest rank that is experimented on. As seen from fig. 2a we do however notice that CLIP is better

than the other, and can have significant improvements both in terms of accuracy as well as fairness metrics with LoRA. At the same time, it is also worth noting that in fig. 2b, we can see that LoRA models may or may not have comparable performance when it comes to fairness metrics such as variance of per class loss. Following on this direction, we aim to improve the fairness of a model on downstream task, akin to Das et al. (2024). One of the key distinction from (Das et al., 2024) is that we aim to improve fairness by introducing a fairness constraint that aims to improve the performance of all classes in a dataset as opposed to relying on a fairness specific dataset that can only be used for a small set of bias mitigation usecases.

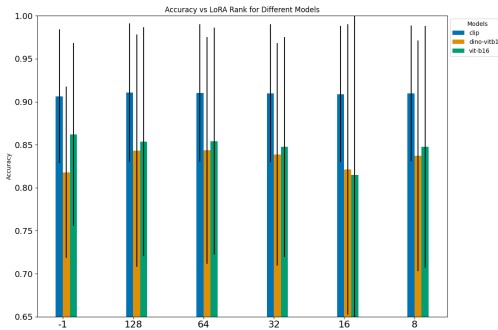 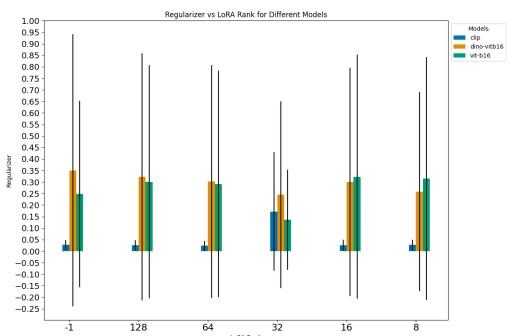

(a) The overall accuracy is (higher) better across models and ranks with LoRA.

(b) The variance of per group loss, varies across models and ranks. Lower value is better.

Figure 2: Comparison of model performance and fairness on the GeoDE across different LoRA ranks as well as FFT. Please note that this doesn't include any specific fairness related intervention. **Rank = -1 implies Full-Finetuning**.

### 4.1 IMPROVING PER CLASS PERFORMANCE

In this work we focus on accuracy parity as a notion of fairness. In order to achieve accuracy parity, the aim is to have equal accuracy across all groups. This is challenging and often times impossible to achieve given the quantized nature of accuracy, number of samples in a group, and difficulty of the samples in each class. Considering the impracticality of directly enforcing accuracy parity, we aim to use a method that still helps to improve the per group accuracy. We introduce a fairness regularizer aimed to reduce the variance of per group loss at a mini-batch level, thereby implicitly improving the performance of under performing group. It is important to note that a degenerate solution to this problem could be by making the performance per class equal, but extremely low–this does not occur here as we still have the original objective that pushes to improve the overall performance of the model and we do not observe this degenerate solution in any of our experimental settings.

In this problem, we aim to minimize the empirical risk over data points $x$, labels $y$, and model parameters $\theta$. Specifically, let the model parameters be $\theta$, which can represent either the entire set of model parameters or the Low-Rank Adaptation (LoRA) parameters, denoted as $\theta_{\text{LoRA}}$, depending on the adaptation approach used in the model. We aim to minimize the total loss function by searching over the parameter space $\Theta$, which includes both the full model parameters and the LoRA parameters:

$$\min_{\theta \in \Theta} \mathcal{J}(\theta) = \min_{\theta \in \Theta} \left( \mathcal{L}(\theta) + \lambda \sum_{g \in \mathcal{G}} \left( \mathcal{L}_g(\theta) - \frac{1}{|\mathcal{G}|} \sum_{g' \in \mathcal{G}} \mathcal{L}_{g'}(\theta) \right)^2 \right) \tag{1}$$

We consider a set of groups $\mathcal{G}$, with each group denoted by $g \in \mathcal{G}$. **Here, the groups can either be the respective classes present in the classification dataset or they can be the sensitive labels associated with each of the sample.** The empirical risk over all data points is represented by $\mathcal{L}(\theta)$, which is typically defined as the average loss over the entire dataset. The second term is a regularization component that penalizes the variance of the average loss across different groups. For each group $g$, the average loss is denoted by $\mathcal{L}_g(\theta)$. Additionally, $\lambda$ is a hyperparameter that controls the strength of the regularization term.

This formulation aims to not only minimize the overall loss but also promote fairness by reducing disparities in performance across different groups. It is important to note that we chose this formulation as a matter of scope, there are other ways of ensuring similar outcomes.

## 5 MEASURING FAIRNESS

We focus on five key metrics to provide a comprehensive assessment of both the model's performance and its fairness. **Here, groups refer to the classes in the dataset and sensitive groups refer to the sensitive labels associated with the classes** These metrics are:

1. **Aggregate Evaluation Accuracy (Acc)**: The overall accuracy of the model across the entire dataset.

2. **Minimum F1 Score Across Groups** ($\min_{g \in G} \mathbf{F1}_g$): The lowest F1 score among all groups $G$, highlighting the worst-performing group and helping to identify significant disparities.

3. **Minimum Recall Across Groups** ($\min_{g \in G} \mathbf{Recall}_g$): The lowest Recall score among all groups $G$, highlighting the worst-performing group and helping to identify significant disparities in terms of misclassifications.

4. **Sensitive Image Accuracy (if applicable, $\mathbf{Acc_{Sensitive}}$)**: The accuracy specifically on sensitive groups, applicable when the dataset contains sensitive labels. This metric measures any spurious correlations or privacy violation with respect to the model.

5. **Difference of F1 Scores Between Worst and Best Groups** ($\mathbf{\Delta F1} = \max_{g \in G} \mathbf{F1}_g - \min_{g \in G} \mathbf{F1}_g$): The gap between the highest and lowest F1 scores across groups, serving as an indicator of fairness by measuring performance disparity.

Additionally, we present results for conventional fairness metrics such as the **Equalized Opportunity Difference** Hardt et al. (2016); Verma & Rubin (2018); Mehrabi et al. (2021), which measures the difference in true positive rates between groups:

$$\text{Equalized Opportunity Difference} = \left| \mathbb{P}(\hat{Y} = 1 \mid Y = 1, S = s_1) - \mathbb{P}(\hat{Y} = 1 \mid Y = 1, S = s_2) \right|$$

where $Y$ is the true label, $\hat{Y}$ is the predicted label, and $S$ is the sensitive attribute, with $s_1$ and $s_2$ being different groups within $S$.

**Equalized Opportunity Difference for Multiple Sensitive Groups.** For multiple sensitive groups, we generalize the Equalized Opportunity Difference (EOD) using a one-vs-all approach. Let $S \in \{s_1, s_2, \ldots, s_k\}$ be the sensitive attribute, and $\hat{Y}$ the predicted outcome. The EOD between group $S = s_i$ and others $S \neq s_i$ is defined as:

$$\text{EOD}_{s_i} = \left| \mathbb{P}(\hat{Y} = 1 \mid Y = 1, S = s_i) - \mathbb{P}(\hat{Y} = 1 \mid Y = 1, S \neq s_i) \right|$$

**Maximal Equalized Opportunity Difference.** We define the maximal EOD, $\text{EOD}_{\text{max}}$, as the maximum disparity across all sensitive groups:

$$\text{EOD}_{\text{max}} = \max_{i \in \{1,2,\ldots,k\}} \left( \text{EOD}_{s_i} \right) \tag{2}$$

This captures the worst-case violation of equalized opportunity across groups and is a key metric for measuring fairness with multiple sensitive attributes.

These metrics collectively provide a well-rounded evaluation of both model performance and fairness across diverse groups.

# 6 EXPERIMENTS

In this section, we present an empirical comparison addressing the various research questions highlighted earlier. The primary goal of our experiments is to fine-tune models using LoRA with minimal disparity. Although reducing disparity may introduce a trade-off with aggregate performance, our aim is to achieve overall accuracy comparable to mitigation-agnostic methods, both with and without LoRA. All models, unless mentioned otherwise are chosen based on the best evaluation accuracy.

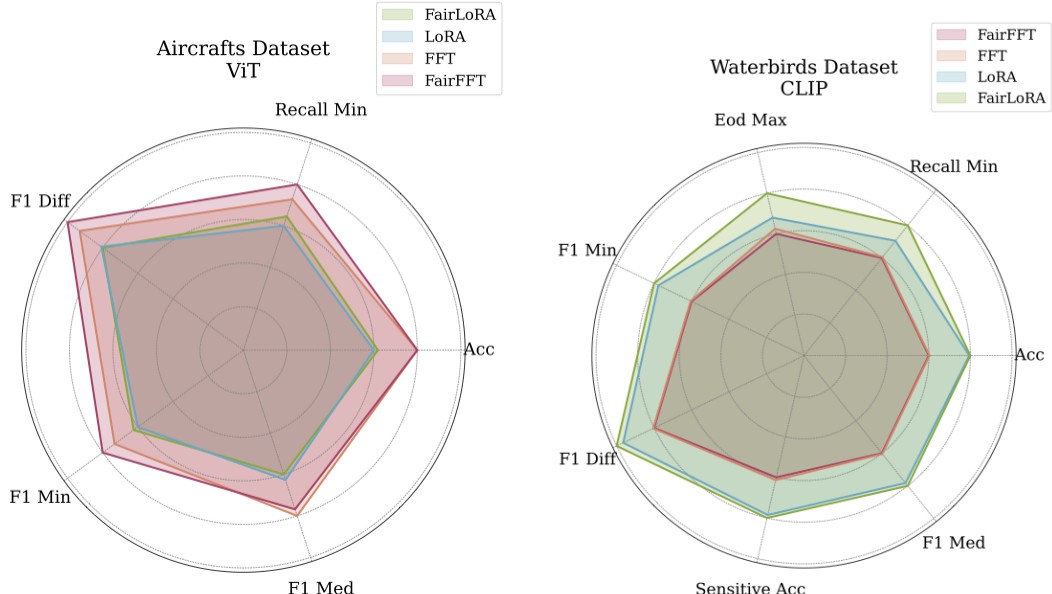

(a) Model: Clip. All metrics are normalized to the same scale and adjusted such that higher is better. In this model, dataset combination, we can notice a dominant behavior with respect to **FairLoRA**.

(b) Model: Clip. All metrics are normalized to the same scale and adjusted such that higher is better. We notice a dominant pattern for FairLoRA across metrics.

Figure 3: Comparison of FairLoRA performance in Clip model across Aircrafts and Waterbirds datasets.

## 6.1 EXPERIMENTAL SETUP

**Tasks and architectures.** We conduct image classification experiments on the Aircrafts Maji et al. (2013), GeoDE Ramaswamy et al. (2024), and Waterbirds Sagawa et al. (2019) datasets. These datasets were chosen because they differ significantly from popular pre-training datasets, as noted in previous works. To provide sufficient variation in architecture, pre-training data, and strategy, we use CLIP Radford et al. (2021), DiNO Caron et al. (2021), and ViT Dosovitskiy (2020) models in our experiments. *All tables, figures unless mentioned otherwise is reported across 3 seeds.* For LoRA as well as FairLoRA only the low rank parameters are updated.

**Vision Models:** CLIP-b32, DINO-b16, and ViT-b16 are key models in computer vision with distinct advantages. CLIP (Contrastive Language-Image Pretraining) excels in cross-modal tasks by learning from large scaled paired image-text data and DINO (Self-Distillation with No Labels) uses self-supervised learning, ideal for tasks without labeled data. Additionally, the commonly available versions also have different pre-training data - both DiNO and ViT are pre-trained on ImageNet Deng et al. (2009), while CLIP is pre-trained on LAION-5B Schuhmann et al. (2022).

**Datasets.** Consistent with prior studies, we use 6,667 training samples and 3,333 test samples from the Aircrafts dataset to perform image classification across 100 classes, noting the high intra-class similarity present in this dataset. For Waterbirds, we perform image classification over 2('landbird' and 'waterbird') classes using 4,795 training samples, 2,400 validation samples, and 2,800 test samples. The per-class distribution of Waterbirds varies across these splits; more details can be found

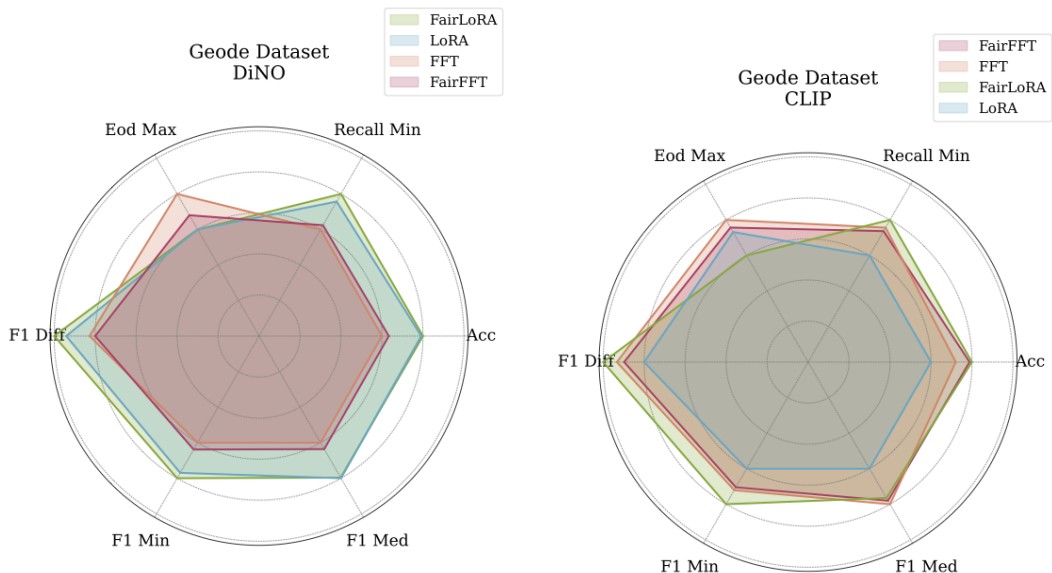

(a) Model: DiNO. We notice a dominant pattern for FairLoRA across metrics apart from EOD, where it is comparable to LoRA

(b) Model: Clip. We notice a dominant pattern for FairLoRA across metrics.

Figure 4: All metrics are normalized to the same scale and adjusted such that higher is better. Comparison of FairLoRA performance on GeoDE across metrics on CLIP and DiNO.

in Sagawa et al. (2019); Pezeshki et al. (2023). Additionally, we utilize the 'land' and 'water' labels as sensitive attributes. These are the classification targets for sensitive image classification. For GeoDE, unlike previous work that uses it solely as an evaluation dataset, we employ it for both fine-tuning and evaluation by performing an 80:20 split on the data. The classification task spans 40 classes, with the 6 geo-location labels used as sensitive attributes.

**Baseline methods.** Our baselines vary depending on the specific research question. Generally, the baselines include LoRA fine-tuning without a fairness regularizer, and full fine-tuning both with and without a fairness regularizer. All models undergo independent pre-training with a comprehensive hyperparameter search.

**Choice of LoRA rank.** For extremely low ranks (less than 4), we observed a significant drop in performance across many experiments. In contrast, performance remained robust across models when using ranks of 8 and above. Consequently, most of our experiments focus on the rank range [8, 128]. We also explore some experiments with extremely low ranks; further details can be found in the appendix.

**Choice of $\lambda$.** We carry out thorough search over potential $\lambda$ values in the range of [0.01, 100], in multiples of 10 independently on all model, dataset, method combinations.

### 6.2 EMPIRICAL EVALUATIONS

**How does FairLoRA improve the fairness?** We see that for most of the experiments, FairLoRA is comparable or better than doing fair full fine-tuning. In particular, in fig. 8a, we can see that Fair-LoRA performs better on multiple metrics. It is important to note that sometimes this improvement in fairness comes at a small cost of aggregate accuracy, but as seen in table 1 and table 2, this is dependent on the underlying base model. Furthermore, we also note that in most of our experiments, FairLoRA performs better across metrics (both in terms of fairness and performance) than LoRA.

**Does the rank have a universal impact in FairLoRA?** In our experiments we notice that, there is no strict trend as to when a higher rank would be required. We see that based on the model, pre-training data and pre-training strategy, the rank required to get a fair model with good overall performance would vary. This can be seen in fig. 5, where the CLIP models get a fair and well performing model for much lower ranks in FairLoRA. It is also important to note that there is no

| Model | Method | Accuracy (↑) | F1 Min (↑) | Recall Min (↑) | Δ F1 (↓) |
|---|---|---|---|---|---|
| | LoRA | 97.35 ± 0.17 | 87.24 ± 0.25 | 83.74 ± 2.15 | 12.29 ± 0.16 |
| CLiP | FairLoRA | **97.58 ± 0.06** | **88.93 ± 0.48** | **86.51 ± 0.17** | **10.71 ± 0.48** |
| | FFT | 97.49 ± 0.19 | 88.26 ± 1.12 | 85.91 ± 0.47 | 11.24 ± 0.92 |
| | FairFFT | 97.57 ± 0.03 | 88.12 ± 0.42 | 85.64 ± 0.47 | 11.52 ± 0.42 |
| | LoRA | 94.38 ± 0.23 | 86.96 ± 0.91 | 82.54 ± 2.61 | 12.93 ± 0.73 |
| DiNO | FairLoRA | **94.53 ± 0.07** | **87.65 ± 0.83** | **83.88 ± 0.83** | **11.99 ± 0.83** |
| | FFT | 91.05 ± 0.84 | 83.08 ± 0.48 | 77.65 ± 2.00 | 14.77 ± 0.10 |
| | FairFFT | 91.63 ± 0.98 | 83.96 ± 1.00 | 78.39 ± 4.36 | 15.20 ± 0.59 |
| | LoRA | 94.29 ± 0.07 | 86.76 ± 0.77 | 83.46 ± 1.67 | 12.84 ± 0.32 |
| ViT | FairLoRA | *94.71 ± 0.08* | *87.09 ± 1.45* | *83.71 ± 2.14* | 12.81 ± 1.26 |
| | FFT | 94.39 ± 0.33 | 87.03 ± 0.32 | 83.45 ± 0.46 | 12.61 ± 0.04 |
| | FairFFT | **94.89 ± 0.27** | **87.44 ± 0.73** | **85.64 ± 0.94** | **12.05 ± 0.40** |

Table 1: The table compares FFT vs LoRA and FairFFT vs FairLoRA for GeoDe. Metrics include: Accuracy, the mean classification accuracy; F1 Min, the minimum F1 score across classes; Recall Min, the minimum Recall across classes; Δ F1, the difference between the maximum and minimum F1 score across classes.

| Model | Method | Accuracy (↑) | F1 Min (↑) | Recall Min (↑) | Δ F1 (↓) |
|---|---|---|---|---|---|
| | LoRA | 93.83 ± 0.81 | 72.74 ± 2.94 | 74.06 ± 1.36 | 19.28 ± 1.84 |
| CLiP | FairLoRA | **93.87 ± 0.58** | **73.45 ± 1.96** | **76.32 ± 0.38** | **18.58 ± 1.15** |
| | FFT | 92.27 ± 1.50 | 67.37 ± 5.80 | 71.68 ± 4.76 | 22.50 ± 3.78 |
| | FairFFT | 92.24 ± 1.38 | 67.23 ± 5.36 | 71.55 ± 4.55 | 22.60 ± 3.52 |
| | LoRA | **89.27 ± 1.12** | **77.13 ± 2.00** | **81.45 ± 0.95** | **15.86 ± 1.24** |
| DiNO | FairLoRA | 89.21 ± 1.09 | 77.01 ± 1.93 | 81.33 ± 0.78 | 15.94 ± 1.18 |
| | FFT | 83.07 ± 1.86 | 64.94 ± 2.91 | 70.55 ± 3.02 | 23.90 ± 1.56 |
| | FairFFT | 82.90 ± 1.04 | 64.36 ± 1.85 | 69.55 ± 1.36 | 24.39 ± 1.14 |
| | LoRA | **91.83 ± 0.08** | **82.05 ± 0.21** | 84.21 ± 0.99 | **12.66 ± 0.15** |
| ViT | FairLoRA | 90.91 ± 1.08 | 80.50 ± 2.23 | **84.59 ± 2.71** | 13.57 ± 1.51 |
| | FFT | 90.02 ± 0.67 | 77.93 ± 1.48 | 79.45 ± 1.70 | 15.62 ± 1.04 |
| | FairFFT | 88.96 ± 0.63 | 76.05 ± 0.87 | 78.95 ± 0.99 | 16.78 ± 0.42 |

Table 2: The table compares FFT vs LoRA and FairFFT vs FairLoRA for Waterbirds. Metrics include: Accuracy, the mean classification accuracy; F1 Min, the minimum F1 score across classes; Recall Min, the minimum Recall across classes; Δ F1, the difference between the maximum and minimum F1 score across classes.

monotonic pattern associated with LoRA ranks when . For most of the experiments even low ranks were comparable both in terms of fairness and performance, and this is something not observed in the previous work on LLMs Das et al. (2024).

**How does FairLoRA handle distribution shifts from the pre-training data?** Based on FID scores Parmar et al. (2022), GeoDE and Waterbirds are farther from both the pre-training distributions(ref table 4 for FID). From table 1, it is clear that FairLoRA is best across methods and with ViTs, it is second only to Fair FFT despite having less than 1% of trainable parameters in comparison. It is also important to note that in table 2, FairLoRA is better than FairFFT and FFT across all metrics and gets comparable performance to LoRA.

**Does FairLoRA perform the same across architectures** Although FairLoRA improves fairness across metrics on most tasks, it is important to note the variance across architectures. In general, we notice that the CLIP models are more adaptable to the FairLoRA and exhibit improvements across all metrics. It is also important to note that CLIP models are almost twice as large as the other models. Furthermore, it is important to note that DiNO models seem to adapt worse to the LoRA based fairness regularization, especially with distribution shift(table 1. ViTs seem to adapt least when the task involves groups with high intra-class similarities and require higher model capacity to improve on both performance and fairness.( table 6)

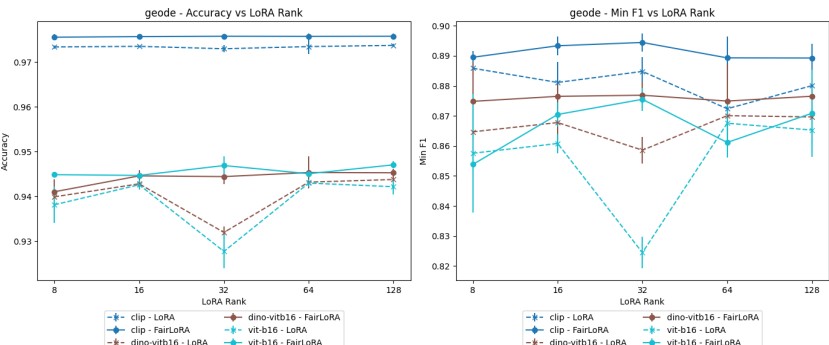

Figure 5: Comparison on the impact of rank on performance as well as fairness across models for GeoDe. Higher value is better in both the graphs. There is no monotonic behaviour on fairness or performance when changing the ranks. FairLoRA versions are more stable to changes in ranks.

**How does FairLoRA affect privacy** The sensitive image accuracy aims to determine how much spurious information is leaked, when we are fine-tuning the model on downstream datasets. Usually, we see that despite not being part of the learning objective models tend to pick up these sensitive information from the data and is often exacerbated when optimized for fairness Fioretto et al. (2022). We can see that in table 3, LoRA models have a lower (better) sensitive image accuracy compared to full finetuning. We also see the similar trend reflected in FairLoRA vs FairFFT, thereby highlighting that forcing fairness doesn't come at the cost of a privacy violation in this setup. Furthermore, we can hypothesise that the low rank matrices work in ways similar to sparse gradients and therefore provide some implicit differential privacy benefits Ghazi et al. (2024); Yang et al. (2023); Malekmohammadi & Farnadi (2024).

**How does FairLoRA perform across the metrics** From fig. 3 it is clear that FairLoRA has a dominant performance across metrics. It is also worth noting that, despite not directly optimizing for it, the regularizer also helps to improve on fairness aspects such as EOD and F1. It is also worth highlighting that in fig. 4, although better in most metrics, FairLoRA tends to be slightly worse on EOD compared to other methods, thereby highlighting the importance of our proposed multi faceted evaluations.

| Model | Method | EOD_max ($\downarrow$) | Sensitive Acc ($\downarrow$) |
|---|---|---|---|
| CLiP | LoRA | 43.86 ± 1.89 | 61.27 ± 1.54 |
| | FairLoRA | **37.84 ± 0.43** | **61.02 ± 1.11** |
| | FFT | 46.62 ± 9.83 | 64.16 ± 3.14 |
| | FairFFT | 47.87 ± 7.68 | 64.39 ± 2.61 |
| DiNO | LoRA | **31.58 ± 0.75** | **60.11 ± 0.97** |
| | FairLoRA | 31.83 ± 0.43 | 60.16 ± 0.93 |
| | FFT | 52.88 ± 5.33 | 66.19 ± 1.77 |
| | FairFFT | 52.38 ± 4.41 | 66.03 ± 1.36 |
| ViT | LoRA | 28.07 ± 1.57 | 57.71 ± 0.25 |
| | FairLoRA | **26.82 ± 5.69** | **58.63 ± 1.18** |
| | FFT | 37.59 ± 2.71 | 59.58 ± 0.63 |
| | FairFFT | 39.10 ± 1.99 | 60.58 ± 0.77 |

Table 3: The table compares sensitive accuracy and the $EOD_{max}$ across models and methods. We can see that FairLoRA is better or comparable to other metrics when we measure on these metrics. Aggregated results for waterbirds. Each metric is aggregated across 3 seeds.

# 7 DISCUSSION

Given the ubiquitous use, it is important to develop parameter-efficient techniques that reliably mitigate fairness issues. While developing such solutions it is important to focus on (i) trade offs in terms of different metrics, and compute and (ii) if the method truly generalizes

## 7.1 TRADE-OFFS

In our experiments, we observed trade-offs between overall performance and fairness metrics when applying FairLoRA. Incorporating the fairness regularizer often led to improved performance on underrepresented groups at the expense of slight reductions in aggregate accuracy. This trade-off is expected, as the regularizer aims to reduce the variance in loss across groups, thereby focusing the model's learning capacity on groups that are harder to predict accurately. *But it is important to note that under most settings, the aggregate accuracy improved or was comparable to that of full-finetuning.*

The choice of the LoRA rank also plays a pivotal role in balancing the trade off between various metrics and computational cost. However, the observations we have show that the effect of rank is not universal and would vary across models, datasets and even metrics - **this is contrary to what was observed by Das et al. (2024).**

## 7.2 GENERALIZATION

The generalization of fairness improvements across different models and datasets is a critical consideration. Our results indicate that although useful, the effectiveness of FairLoRA in mitigating fairness issues is not universal but depends on factors such as the pre-trained model architecture, the nature of the pre-training data, and the specific downstream task. For instance, models such as CLIP, which are pre-trained on diverse multi-modal data, may require lower LoRA ranks to achieve fairness compared to models pre-trained on more homogeneous datasets.

Moreover, the distribution shift between pre-training and fine-tuning datasets as well as the intra-class similarities within the fine-tuning dataset can influence the model's ability to generalize fairness improvements. **Although this was hypothesized in Ding et al. (2024); Das et al. (2024), we offer a more comprehensive empirical evaluation in this regard**

# 8 CONCLUSION

In this work, we introduced FairLoRA, a fairness-aware LoRA approach by incorporating a fairness regularizer aimed at reducing the variance of per-group loss, thereby improving performance on underrepresented groups. Through comprehensive experiments across various models (CLIP, DINO, ViT), datasets (Aircrafts, GeoDE, Waterbirds), and fairness metrics, we found that LoRA does not introduce systemic disparate impact and FairLoRA can achieve fairness outcomes comparable to or better than Fair full fine-tuning (FFT).

Our findings highlight the importance of evaluating multiple fairness metrics to capture a holistic view of a model's performance and fairness implications. We observed that there is no universal trend with respect to LoRA ranks; the optimal rank depends on the specific model, pre-training data, and task. Additionally, we examined the effects of distribution shifts between pre-training and fine-tuning datasets, and notice how efficiently FairLoRA can adapt.

Overall, our study demonstrates that FairLoRA is a viable and efficient alternative to FFT for mitigating fairness issues in machine learning models. Future work could extend this analysis to other architectures, datasets, and definitions of fairness, as well as explore intersectional fairness. **More importantly, it would be interesting to study the impact of FairLoRA on image segmentation tasks, where we see long-tailed distributions.**

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
