# A    APPENDIX

## A.1    REPRODUCIBILITY STATEMENT

All our code and model weights will be open-sourced post the conference anonymity period and will be available to the larger community to use under open source licensing.

## A.2    EQUAL OPPORTUNITY IN FAIRNESS

Equal Opportunity ensures that the **True Positive Rates (TPR)** are equal across demographic groups. Mathematically:

$$\text{TPR}_{\text{Group 1}} = \text{TPR}_{\text{Group 2}} = \ldots$$

where:

$$\text{TPR} = \frac{\text{True Positives (TP)}}{\text{True Positives (TP)} + \text{False Negatives (FN)}}$$

### EXAMPLE: LOAN APPROVAL

Consider a loan model evaluating two demographic groups with the following outcomes:

| Group | True Positives (TP) | False Negatives (FN) | TPR |
|-------|---------------------|----------------------|-----|
| A | 80 | 20 | $\frac{80}{80+20} = 0.80$ |
| B | 60 | 40 | $\frac{60}{60+40} = 0.60$ |

The model violates Equal Opportunity because $\text{TPR}_A \neq \text{TPR}_B$.

### FIX: THRESHOLD ADJUSTMENT

Adjusting decision thresholds can equalize the TPR:

- Group A: Keep the threshold at $0.5$.
- Group B: Lower the threshold to $0.4$.

After adjustment, the outcomes are:

| Group | True Positives (TP) | False Negatives (FN) | TPR |
|-------|---------------------|----------------------|-----|
| A | 80 | 20 | 0.80 |
| B | 72 | 28 | $\frac{72}{72+28} = 0.80$ |

Now $\text{TPR}_A = \text{TPR}_B = 0.80$, satisfying Equal Opportunity.

### FAIRNESS CONSTRAINT

During training, Equal Opportunity can be enforced as:

$$\left|\text{TPR}_{\text{Group A}} - \text{TPR}_{\text{Group B}}\right| \leq \epsilon$$

or by adding a penalty to the loss function:

$$\mathcal{L}_{\text{fair}} = \mathcal{L}_{\text{original}} + \lambda \cdot \left|\text{TPR}_{\text{Group A}} - \text{TPR}_{\text{Group B}}\right|$$

where $\lambda$ controls the trade-off between fairness and accuracy.

## A.3    FRÉCHET INCEPTION DISTANCE (FID) SCORES - DISTRIBUTION SHIFT

The Fréchet Inception Distance (FID) is a widely-used metric to evaluate the quality of generated images by comparing them to real images. It works by calculating the distance between the feature distributions of two datasets (real and generated images) using the activations of the InceptionV3

model. Specifically, the FID score is computed using the mean and covariance of these activations, assuming a multivariate Gaussian distribution.

In our setup, we employed the Clean FID library Parmar et al. (2022) to calculate the FID between various datasets. For large datasets like LAION-5B, we sampled random subsets and computed the FID on these smaller samples to manage computational constraints. For smaller datasets such as Waterbirds, GeoDe, and Aircrafts, we used the entire dataset for the calculation.

| Fine-Tuning Data | ImageNet | LAION |
|---|---|---|
| Aircrafts | 181.98 | 229.72 |
| Waterbirds | 117.61 | 142.46 |
| GeoDE | 54.38 | 56.91 |

Table 4: FID comparison between the fine-tuning and pre-training data.

## A.4 TRAINABLE PARAMETER RATIOS WITH LORA/FAIRLORA

The number of trainable parameter remain the same for LoRA as well as FairLoRA for the same rank. All models have similar ratios for % of Trainable parameters.

| Model | Rank | Trainable Params | Total Params | (% of Trainable) |
|---|---|---|---|---|
| | 8 | 325,672 | 86,155,088 | 0.38 |
| | 16 | 666,724 | 86,155,088 | 0.77 |
| DiNO | 32 | 1,256,548 | 86,155,088 | 1.44 |
| | 64 | 2,436,196 | 86,155,088 | 2.76 |
| | 128 | 4,795,492 | 86,155,088 | 5.29 |

Table 5: Summary of model parameters for DiNO. The table includes the rank, number of trainable parameters, total parameters, and the percentage of trainable parameters.

## A.5 GRADIENT UPDATES IN LORA

The gradient updates in LoRA apply only to the low-rank matrices $A$ and $B$, while the pre-trained parameters $\theta_0$ remain fixed. Given an objective function $\mathcal{L}$, the gradients of the loss with respect to $A$ and $B$ are computed as:

$$\frac{\partial \mathcal{L}}{\partial A} = \frac{\partial \mathcal{L}}{\partial \theta} B^\top, \quad \frac{\partial \mathcal{L}}{\partial B} = A^\top \frac{\partial \mathcal{L}}{\partial \theta}$$

Here, $\frac{\partial \mathcal{L}}{\partial \theta}$ is the gradient of the loss with respect to the full parameter matrix $\theta$. These updates allow the model to adapt to the downstream task with far fewer trainable parameters, preserving most of the pre-trained knowledge while fine-tuning for the new task.

LoRA's parameterization is particularly effective in large models where the parameter matrices are high-dimensional, as it avoids the computational cost of updating the entire matrix. By focusing on the low-rank updates, LoRA achieves a balance between fine-tuning flexibility and resource efficiency.

For further details, the original LoRA formulation and its theoretical justification can be found in Hu et al. (2021).

## A.6 GRADIENTS IN FAIRLORA

Let $\theta = \theta_0 + AB$, where $\Delta\theta = AB$ is the LoRA low-rank update. We need to compute the gradients of $\mathcal{J}(\theta)$ with respect to both $A$ and $B$.

GRADIENT WITH RESPECT TO $A$:

$$\frac{\partial \mathcal{J}}{\partial A} = \frac{\partial \mathcal{L}}{\partial \theta} B^\top + \lambda \sum_{g \in \mathcal{G}} 2 \left( \mathcal{L}_g(\theta) - \frac{1}{|\mathcal{G}|} \sum_{g' \in \mathcal{G}} \mathcal{L}_{g'}(\theta) \right) \frac{\partial \mathcal{L}_g(\theta)}{\partial \theta} B^\top$$

GRADIENT WITH RESPECT TO $B$:

$$\frac{\partial \mathcal{J}}{\partial B} = A^\top \frac{\partial \mathcal{L}}{\partial \theta} + \lambda \sum_{g \in \mathcal{G}} 2 \left( \mathcal{L}_g(\theta) - \frac{1}{|\mathcal{G}|} \sum_{g' \in \mathcal{G}} \mathcal{L}_{g'}(\theta) \right) A^\top \frac{\partial \mathcal{L}_g(\theta)}{\partial \theta}$$

## A.7 EMPIRICAL EVALUATIONS CONTINUED

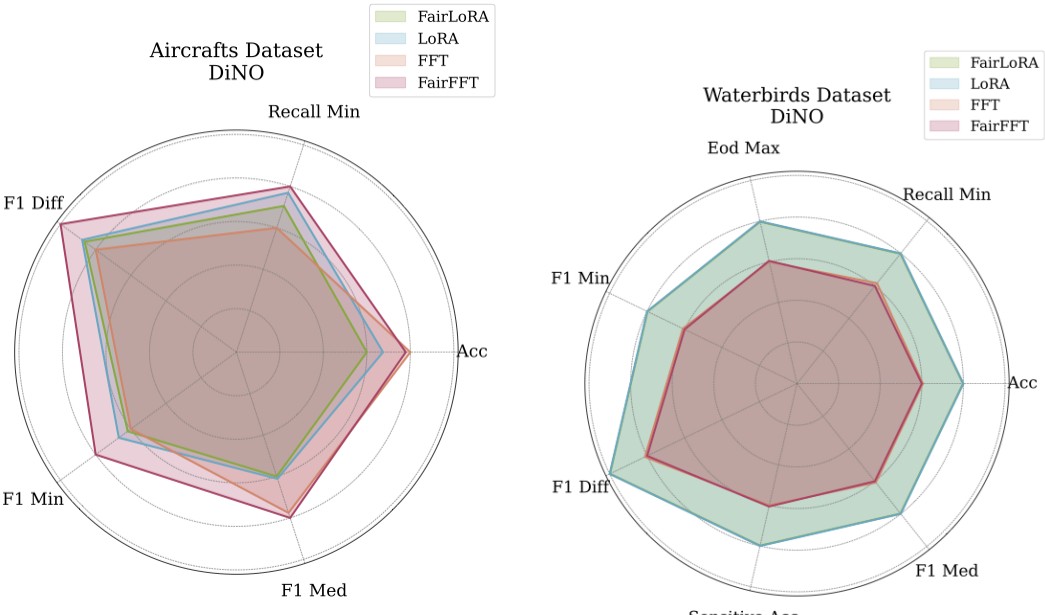

(a) Model: DiNO. We notice a dominant pattern for FairLoRA across metrics apart from EOD, where it is comparable to LoRA

(b) Model: DiNO. We notice a dominant pattern for FairLoRA across metrics.

Figure 6: All metrics are normalized to the same scale and adjusted such that higher is better. Comparison of FairLoRA performance on DiNO model across datasets.

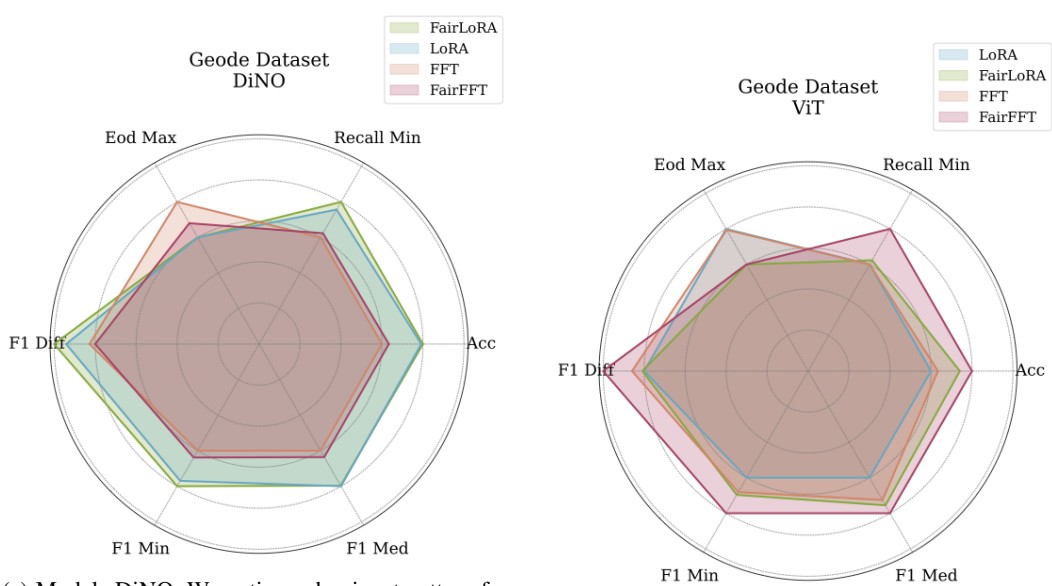

(a) Model: DiNO. We notice a dominant pattern for FairLoRA across metrics apart from EOD, where it is comparable to LoRA

(b) Model: ViT. We notice a dominant pattern for FairFFT across metrics.

Figure 7: All metrics are normalized to the same scale and adjusted such that higher is better. Comparison of FairLoRA performance on GeoDE across metrics on ViT and DiNO.

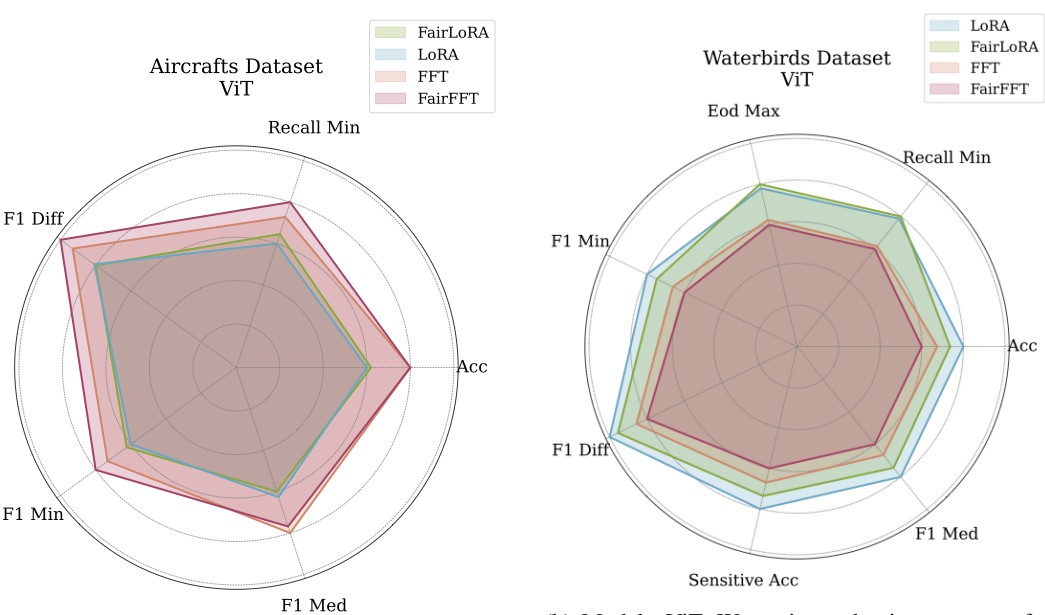

(a) Model: ViT. We notice a dominant pattern for FairFFT

(b) Model: ViT. We notice a dominant pattern for LoRA across metrics but FairLoRA achieves similar scores.

Figure 8: All metrics are normalized to the same scale and adjusted such that higher is better. Comparison of FairLoRA performance on ViT model across datasets.

| Model | Method | Accuracy (↑) | F1 Min (↑) | Recall Min (↑) | Δ F1 (↓) |
|-------|--------|--------------|------------|----------------|----------|
| CLiP | LoRA | 81.98 ± 1.38 | 17.60 ± 5.05 | 14.97 ± 5.10 | 80.92 ± 5.00 |
| | FairLoRA | 86.67 ± 2.56 | **27.44 ± 15.30** | **24.81 ± 14.07** | **71.06 ± 13.78** |
| | FFT | 81.92 ± 1.21 | 21.28 ± 5.31 | 18.78 ± 3.14 | 77.21 ± 5.29 |
| | FairFFT | **87.03 ± 2.81** | 22.82 ± 12.47 | 17.94 ± 10.30 | 76.17 ± 10.72 |
| DiNO | LoRA | 69.39 ± 0.87 | 21.43 ± 2.81 | 19.99 ± 4.57 | 77.59 ± 1.97 |
| | FairLoRA | 68.28 ± 0.32 | 20.64 ± 4.67 | 18.75 ± 5.92 | 77.88 ± 4.65 |
| | FFT | **71.26 ± 0.06** | 20.37 ± 7.65 | 16.67 ± 6.79 | 79.15 ± 6.81 |
| | FairFFT | 70.95 ± 0.95 | **23.46 ± 0.11** | **20.59 ± 2.94** | **75.10 ± 1.32** |
| ViT | LoRA | 70.06 ± 0.92 | 25.99 ± 3.11 | 24.24 ± 3.03 | 72.03 ± 2.28 |
| | FairLoRA | 70.45 ± 0.48 | 27.11 ± 2.11 | 25.49 ± 3.40 | 72.41 ± 1.27 |
| | FFT | 74.17 ± 0.31 | 32.18 ± 5.83 | 27.75 ± 6.28 | 66.34 ± 5.80 |
| | FairFFT | **74.18 ± 0.64** | **35.36 ± 5.04** | **29.71 ± 2.99** | **63.15 ± 4.99** |

Table 6: The table compares FFT vs LoRA and FairFFT vs FairLoRA for Aircrafts. Metrics include: Accuracy, the mean classification accuracy; F1 Min, the minimum F1 score across classes; Recall Min, the minimum Recall across classes; Δ F1, the difference between the maximum and minimum F1 score across classes.

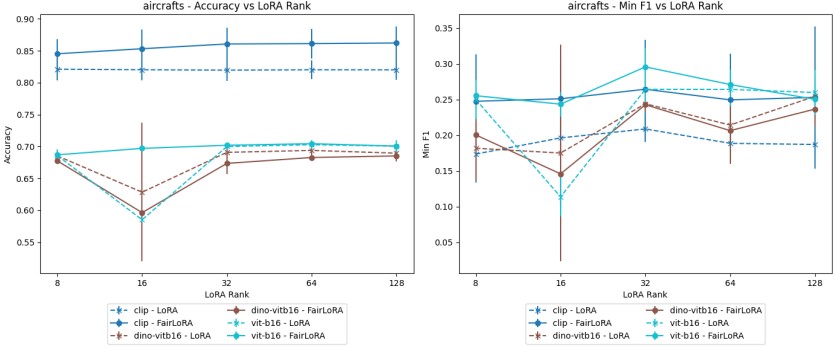

Figure 9: Comparison on the impact of rank on performance as well as fairness across models for Aircrafts. Higher value is better in both the graphs.

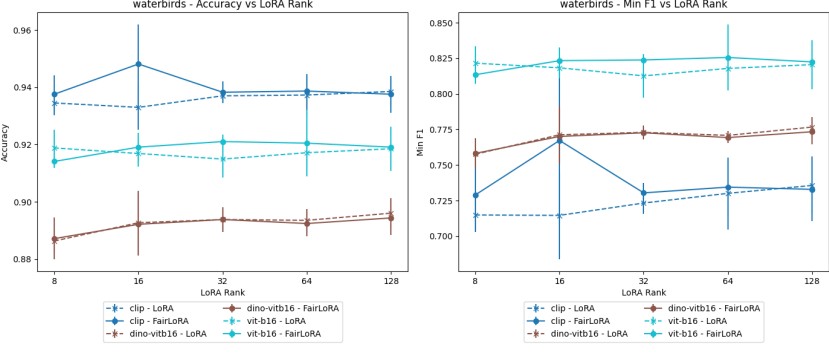

Figure 10: Comparison on the impact of rank on performance as well as fairness across models for Waterbirds. Higher value is better in both the graphs.

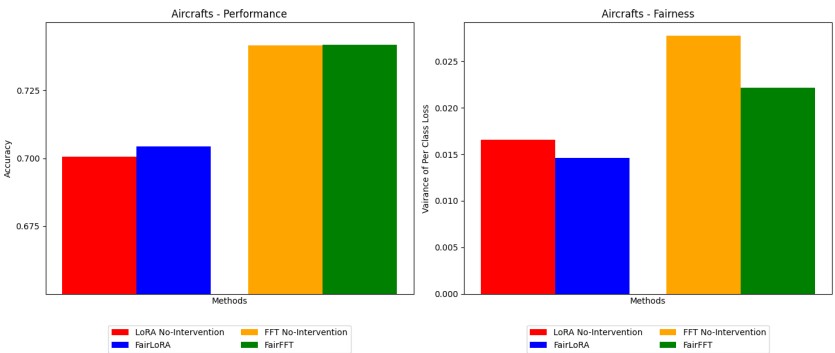

Figure 11: On the left, we plot the performance of the model with and without fairness regularizers for both full finetuning (FFT) as well as LoRA. We notice that using the regularizer improves overall performance as well in this particular example. On the right, we visualize the effect on the variance of loss across classes and notice that this variance is lower for the ones with regularizer and the combination of LoRA and the regularizer yeilds the best results.

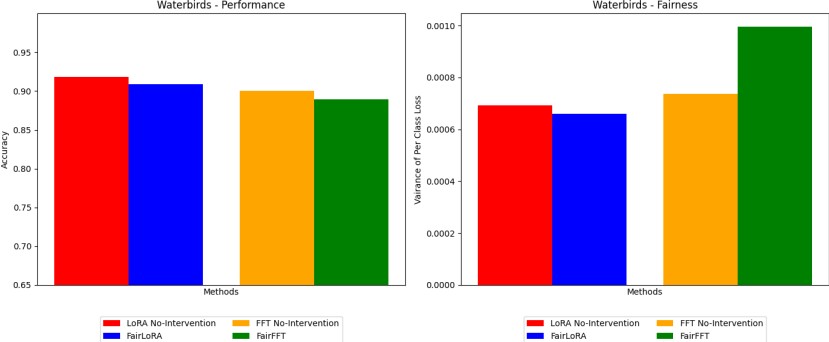

Figure 12: On the left, we plot the performance of the model with and without fairness regularizers for both full finetuning (FFT) as well as LoRA. On the right, we visualize the effect on the variance of loss across classes.