# OpenReview forum: "FairLoRA: Unpacking Bias Mitigation in Vision Models with Fairness-Regularized Low-Rank Adaptation"
_ICLR.cc/2025/Conference — Submitted to ICLR 2025_

### Official Review · Reviewer_Kvf3 · 2024-10-31

**Soundness:** 3
**Presentation:** 3
**Contribution:** 3
**Rating:** 6
**Confidence:** 3

**Summary:**

This paper studies the fairness problem of the popular LoRA (Low-Rank Adaptation) technique. The authors propose a simple yet effective strategy for improving the fairness of LoRA, specifically by reducing the variance of per-group loss. The results are promising.

**Strengths:**

- The topic of fairness in LoRA (Low-Rank Adaptation) is important and worthy of study.
- The proposed method of reducing group-wise variance is simple yet effective.
- The motivation behind the proposed method is clear.
- The method is evaluated on several visual foundation models, and the results are convincing.

**Weaknesses:**

- How are the groups divided in page 4, line 212?
- The formulation in page 4, line 208 lacks a number.
- How are the groups divided in Section 5, "MEASURING FAIRNESS"? What is the difference between these groups and the groups mentioned in page 4, line 212?
- In my opinion, fairness means consistent results among different categories, such as cats and dogs. What is the relationship and difference between the category group and the groups you defined?
- The formulation of "Equalized Opportunity Difference (EOD)" is abstract. Please add a textual explanation.
-  This paper only studies the recognition problem. How does the proposed method generalize to more fine-grained tasks, such as segmentation? Category imbalance is a common issue in segmentation tasks.

**Questions:**

- Explain the "Group" strategies.

- Using text to explain the formulations.

- How about adopt your approach on segmentation task.

---

> ### Author Response · Authors · 2024-11-25
>
> # Response to Reviewer
>
> We would like to thank the reviewer for highlighting the simple yet effective strategy we propose and recognizing the thoroughness of our experiments and the comprehensive analysis that backs the paper’s story. We have addressed the reviewer comments below.
>
> ## Clarification on the Paper
>
> - We have added information about the group divisions and differences in **Section 5 (lines 224-225, bolded text)**. For more definitions about the groups and splitting, please see our response to **Reviewer P37m** on “Clarification on groups.” - https://openreview.net/forum?id=pB3KeBCnQs&noteId=iCPbraS82Q
> - We have also added the explanation of **EOD** with examples, but shifted it to the appendix (**Section A.2, lines 709-752**) to conform to the page limits.
>
> ## Segmentation Problem
>
> We would like to thank the reviewer for pointing out an exciting direction to explore. We agree with the reviewer that it is indeed an interesting venue for learning long-tail distributions. We have updated the **future works section** to highlight how this could be an important aspect to study (**lines 533-535, bolded text**).
>
> Unfortunately, we don’t think it is within the scope of the current paper or rebuttal timelines to explore the direction of image segmentation in our existing pipeline. In particular, because:
> - Segmentation requires a new set of datasets, model backbones, and training pipelines.
> - Although feasible, it is not extremely straightforward to modify our current regularizer for the segmentation problem within the given timelines.
>
> We would like to emphasize that our paper is among the first to address the problem of long-tailed, fairness-relevant datasets in the context of vision models. Our work makes a significant contribution by carefully selecting datasets and models to provide a comprehensive analysis across several dimensions:
>
> 1. **Diverse Data Distributions**: We examine datasets with varying characteristics to capture a broad spectrum of fairness and long-tailed challenges (**Section 6.1**).
> 2. **Generalization Across Pre-training and Fine-tuning**: We analyze how models generalize when pre-trained and fine-tuned on different datasets (i.e., under distribution shifts) (**Section 6.2 and 7**).
> 3. **Impact of Pre-training Strategies**: We evaluate the effect of various pre-training methods and architectures (**Section 6.2**).
> 4. **Pre-training Data Variability**: We study the influence of different pre-training datasets on downstream fairness and performance (**Section 7**).
>
> We once again thank the reviewer for their time and valuable feedback. We hope we have addressed all the queries. If so, we kindly request the reviewer to increase the scores they have provided. We would also be happy to engage in further discussions as well.

---

> > ### Comment · Reviewer_Kvf3 · 2024-11-28
> > **Thanks for your response**
> >
> > Most of my concerns have be resolved.

---

> > > ### Author Response · Authors · 2024-12-01
> > >
> > > Dear reviewer Kvf3,
> > > Thank you for taking the time to review our rebuttal. We greatly appreciate your thoughtful feedback, which has helped us clarify and strengthen our work.
> > >
> > > If there are remaining concerns or points you feel are unresolved, please share them with us in the remaining discussion period. We are committed to providing any additional clarification or evidence that may be needed.
> > >
> > > Additionally, if your concerns have been addressed, we kindly encourage you to update your score accordingly. Your engagement is invaluable to us, and we deeply appreciate the constructive dialogue throughout this process.

---

### Official Review · Reviewer_P37m · 2024-11-01

**Soundness:** 3
**Presentation:** 3
**Contribution:** 2
**Rating:** 5
**Confidence:** 4

**Summary:**

This paper introduces a novel fairness-based finetuning technique based on LoRA with a fairness-specific regularizer, studies multiple fairness metrics, and analyzes factors for bias mitigation. Experiments conducted on various vision models including ViT, DiNO, and CLIP support the discovery and analysis.

**Strengths:**

1. This paper is the first to introduce a fairness-based fine-tuning technique using LoRA, as stated.

2. It presents multiple fairness metrics (Sec. 5) and conducts extensive experiments, supplementing previous fairness-centered evaluation metrics and contributing to future research.

3. The comprehensive experiments are visually presented with colorful figures, enhancing clarity and accessibility, especially for researchers who may not be familiar with ML fairness concepts.

4. Using these metrics, the paper also examines potential factors influencing FairLoRA’s fairness, such as LoRA ranks, distribution shifts from pretraining data, and task types, further contributing to the research community.

**Weaknesses:**

1. The analysis on handling distribution shifts seems to be insufficient, as it includes only FID-based quantitative comparisons. Would it be possible to conduct additional experiments, such as t-SNE, to better demonstrate the effectiveness of distribution shift handling? For instance, showing the t-SNE point distributions before and after fine-tuning or comparing t-SNE point distributions of fine-tuned models with vanilla LoRA and FineLoRA could add valuable insights.

2. Several recent papers [1][2] also focus on LoRA fairness, discussing similar concepts, such as fairness metrics like EOD and LoRA ranks as fairness factors. Both papers were submitted in May 2024, more than three months before the ICLR submission deadline, yet they are not mentioned in the discussion. It might be helpful to clarify the distinctions in motivation, methodology, experiments, and findings between Submission 11853 and these related works. For instance, [1] notes that "lower ranks may retain the bias of the original model." Does this finding complement or challenge your conclusions? Since [1][2] evaluate LoRA fairness in LLMs rather than vision-language models like CLIP, could extending your experiments to LLMs provide additional insights?

3. In the Geode dataset, FairLoRA’s "Eod Max" value is significantly lower than that of the other methods (Fig. 4, 7). Are there additional comparisons or analyses to further investigate this? Providing a detailed analysis of why FairLoRA performs exceptionally well on this metric for this dataset, as well as exploring whether this trend holds across different model architectures or LoRA ranks, could strengthen the findings.

[1]	Das S, Romanelli M, Tran C, et al. Low-rank finetuning for LLMs: A fairness perspective[J]. arXiv preprint arXiv:2405.18572, 2024.

[2]	Ding Z, Liu K Z, Peetathawatchai P, et al. On Fairness of Low-Rank Adaptation of Large Models[J]. arXiv preprint arXiv:2405.17512, 2024.

**Questions:**

1. If possible, it would be beneficial to include further comparisons and analyses, particularly visualizations, discussions of concurrent papers, and detailed analysis of the EOD in the Geode datasets, as noted in the Weaknesses section.

2. As I am not an expert in ML fairness, I’m curious about how you divide the datasets into different groups. Are the groups derived from inherent divisions in the dataset, random divisions, or based on the original VLM’s performance on each sample during inference? Could different group division strategies affect the performance and conclusions? It would be helpful if you could clarify the exact process for group division in each dataset and discuss how alternative division strategies might influence your results and conclusions about fairness.

3. In figure 5, why a LoRA Rank of 32 experiences a sudden drop in Accuracy and Min F1, compared with 16 and 64? Similarly, in Figure 9 there is a sudden drop of performance when LoRA rank is set to 16, compared with neighbored rank values. You may further analyze that if this pattern is consistent across different models or datasets, and what implications this might have for choosing optimal LoRA ranks in fairness-aware fine-tuning.

---

> ### Author Response · Authors · 2024-11-25
>
> # Response to Reviewer
>
> We would like to thank the reviewer for highlighting the thoroughness of our experiments and the comprehensive analysis that backs the paper’s story. We have addressed the reviewer comments below.
>
> ## Comparing with Concurrent Work
>
> We would like to highlight that our work is different from **Ding et al., (2024)** as they do not do fine-tuning with fairness in mind. They perform fine-tuning for a downstream task and measure the disparate impact of using LoRA. In our paper, we explicitly focus on fine-tuning for fairness and we measure how this is reflected with and without the fairness regularizer.
>
> As for **Das et al., (2024)**, they focus their analysis on LLMs and it is based on having tailored de-biased data for fine-tuning. Our paper deals with vision models and furthermore, provides a much more scalable approach than having custom datasets for fairness-specific fine-tuning.
>
> The papers highlighted by the reviewer are already discussed in the paper, particularly in the **related works section** as well as **Section 4 (lines 158-169)**. Furthermore, **Fig 1** is also a comparison similar to Ding et al., to motivate the need for “FairLoRA” as opposed to just LoRA.
>
> Although already discussed in our paper, we agree with the reviewer that the discussion of previous work can be expanded in **Section 7**. In particular, the observation of lower ranks being unfair does not hold well in our experiments. As seen in **Fig 5**, FairLoRA gets good performance as well as better fairness even in lower ranks and there is not necessarily a monotonic pattern with the ranks - more details can be found in **Section 6.2 (lines 374-418)**. We have added this discussion in **Sections 7.1 and 7.2** and bolded the relevant text (**lines 503-504** and **515-516**).
>
> ## Further Analysis on LLMs
>
> The reviewer makes a great point and highlights a new direction to explore, but unfortunately, we believe it is not within the scope of this paper. In particular, because the definition of bias and measurement of bias in LLMs are different from vision models. Furthermore, Das et al. use a new “de-biased” dataset to finetune the model while we use an update to the loss function. Given the current state of literature, there is no straightforward way for us to include the current proposed regularizer to an LLM framework.
>
> ## Clarification on Groups
>
> We thank the reviewer for asking this question. We have also improved the definitions and explanations in the original paper, in **Section 5 (lines 224-225, bolded text)**. In short, the different groups are divided as such:
>
> - **GeoDE**: The dataset has 6 classes representing locations, which are often treated as the “sensitive groups" in the literature. The downstream task in this classification problem is to correctly classify an object from one of these 6 different locations to the correct class among 40 objects.
> - **Waterbirds**: The dataset has 2 different classes, landbird and waterbird, for downstream classification. The dataset further has tags associated with land and water, which are generated synthetically. Some of the samples are waterbirds, synthetically placed in a land background. Therefore, the tags about “water” and “land” are often treated as sensitive attributes in the literature. When we measure **EOD_max**, we are measuring the performance disparity on the actual classification task (landbird vs. waterbird) across the different sensitive groups (“land” and “water”).
> - **Aircrafts**: In this dataset, the different classes are the groups themselves. Therefore, we have 100 groups and no sensitive groups; hence we don’t report EOD or Sensitive Image Accuracy in this dataset.
>
> In short, the reviewer is right: the conclusion changes based on what groups we are measuring for. For instance:
> - In **GeoDE**, when we look at **min F1** or **min recall**, we focus on the group among the 40 classes that got the least performance, whereas when we look at **EOD_max**, we are looking at which sensitive group performed the least across all 40 classes. This definitely changes how you interpret the fairness of the task.
>
> ## Further Analysis and Visualizations
>
> We thank the reviewer for highlighting our experiments with respect to distribution shift. As suggested, we are working on the **t-SNE plots** that show the variation in the distributions before fine-tuning and after fine-tuning, comparing both full fine-tuning as well as LoRA/FairLoRA. Please let us know if this is something that aligns with what the reviewer is looking for.

---

> > ### Author Response · Authors · 2024-11-25
> > **(Contd...)**
> >
> > ## Non-Monotonic Patterns in LoRA Ranks
> >
> > The reviewer is right in saying that there is a sudden change in the performance at some ranks. This non-monotonicity is observed in most datasets as well as models. The rank at which this pattern occurs and the magnitude of change might differ, though. We hypothesize this as a **regularizer effect of the LoRA ranks**. Similar observations have been made about LoRA ranks (although not in the context of fairness) having a regularizing effect and how some intermediate ranks can have out-of-the-box performance [2, 3].
> >
> > As the reviewer pointed out, this can be observed in **Fig 5, 9, and 10**, where we see experiments across 3 different models on 3 different datasets across multiple ranks.
> >
> > ---
> >
> > ## References
> >
> > [1] Nwatu, J., Ignat, O., and Mihalcea, R., 2023. Bridging the digital divide: Performance variation across socio-economic factors in vision-language models. *arXiv preprint arXiv:2311.05746*.
> > [2] Hu, E.J., Shen, Y., Wallis, P., Allen-Zhu, Z., Li, Y., Wang, S., Wang, L., and Chen, W., 2021. Lora: Low-rank adaptation of large language models. *arXiv preprint arXiv:2106.09685*.
> > [3] Shuttleworth, R., Andreas, J., Torralba, A., and Sharma, P., 2024. LoRA vs. Full Fine-tuning: An Illusion of Equivalence. *arXiv preprint arXiv:2410.21228*.

---

### Official Review · Reviewer_42sa · 2024-11-03

**Soundness:** 2
**Presentation:** 2
**Contribution:** 2
**Rating:** 3
**Confidence:** 4

**Summary:**

This paper proposes to achieve Lora fairness by reducing the variance of per group loss at a mini-batch level, which can reduce the performance disparities of the Lora across all classes. Experiments on Aircrafts with 100 classes and Waterbirds with 2 classes show that the proposed method is somewhat effective.

**Strengths:**

1. The investigation of fairness on Lora is interesting.

**Weaknesses:**

1. This paper is difficult to read.
- Figures: Fig.1, and 2,5 are not well illustrated, the font is too small to read clearly.  Also, the low variance between losses does not necessarily lead to good performance/fairness.
- Writing: The story is not clearly or well organized. Fairness is not well defined in the classification tasks (which is also very limited) involved in this paper. The fairness can be defined between classes on a specific task, or between different metrics, or between different tasks, or even on different domains, which is not even mentioned in this paper. The distinctions/relationships between fairness, good performance, and good generalization ability are not discussed.
2. Poor experiments.
- Experiments on two datasets are not convincing enough, and the dataset Waterbirds only has two classes, which is not a good choice to demonstrate fairness.
- Fairness may be largely affected by the long-tail distribution of the training data, and there are no experiments. Also, there should be some comparisons between strategies for long-tail problems and the proposed method.
- Experiments only show min and avg metrics, without the best metrics. It is doubted that the best group may be degenerated by the proposed method.
3. Limited novelty.
- Minimizing the group loss variance is straightforward and seems effective, but there should be more explanations, especially on why the results do not degenerate.

**Questions:**

see weaknesses.

---

> ### Author Response · Authors · 2024-11-25
>
> We would like to thank the reviewer for highlighting that our analysis on Fairness in LoRA and FairLoRA are interesting. We would like to respond to the questions the reviewer had below.
>
> ## Low variance of loss doesn’t mean good performance/fairness
>
> We agree with the reviewer that a low variance of loss alone doesn’t guarantee either, but the overall objective also has the ERM terms which makes sure that the model still optimizes for good performance and thereby not having a degenerate solution (poor performance but low variance in losses) - see Equation 1 in section 4.1. The combination of the ERM term with the regularizer that aims to reduce the variance of losses ensures that, apart from overall performance, the model also focuses on improving every group individually.
>
> ## Fairness Definitions
>
> We thank the reviewer for pointing out the definitions of fairness in our paper. We would like to emphasize that one of the main aspects of the paper is to show how different measurements of fairness in classification can have different interpretations. We would also like to bring the reviewer's attention to section 5 where we talk in detail about measuring fairness and all the metrics we use.
>
> As the reviewer correctly observed, “fairness can be defined between classes on a specific task, or between different metrics, or between different tasks, or even on different domains”:
>
> - Metrics such as **min F1 score per class**, **min Recall per class** tend to focus on the fairness between classes.
> - Metrics such as **EOD**, **Sensitive image accuracy** focus on the performance of the model on sensitive groups on the same downstream task.
> - We perform **sensitive attribute prediction** on the Waterbirds dataset, which is a measure of fairness between tasks as the sensitive attribute prediction is not a part of the original downstream task finetuning and we tend to measure the performance disparity in that [1, 2].
>
> Additionally, the EOD calculations on **GeoDE** also show how the performance disparity exists between the same class across different sensitive attributes (locations they are from). For instance, a high EOD_max score implies there are some classes which perform poorly only if their sensitive attributes belong to one of the specific countries. The ΔF1 for FairLoRA with CLiP on Waterbirds is 18.58 +/- 1.15 (ref. Table 2) while the EOD max score is 37.84 +/- 0.43.
>
> This highlights that the difference in performance becomes more visible when the performance is grouped and categorized based on sensitive attributes - this is indeed an interesting example of how different measurements can change the perception of fairness.
>
> ## Dataset Variations and Fairness
>
> Although our analysis is focused on the vision domain, our datasets are well selected to highlight different variations:
>
> - **Aircrafts**: A dataset with high intra-class similarity.
> - **Waterbirds**: A handcrafted dataset aimed to highlight spurious correlations with a skewed distribution in test and balanced in train.
> - **GeoDE**: A dataset with long-tail distribution (when considered for per class per region and the variety/difficulty of samples) [8, 9].
>
> It is a bit unclear what the reviewer means by fairness across domains. It would be helpful if they can share any relevant citations on what they are looking for.
>
> ## Experiments
>
> We would like to respectfully disagree with the reviewer about the poor quality of experiments and insufficiency of the datasets/models combo. This viewpoint is also contradictory to the remaining reviewers who highlight that our experiment section is comprehensive and clear.
>
> We run our experiments on three datasets - **GeoDE** (Table 1, Figure 1, 2, 4, 5, 7), **Aircrafts** (Table 6, Figure 3a, 6a, 8a, 9, 11), and **Waterbirds** (Table 2, 3, Figure 3b, 6b, 8b, 10, 12), with detailed discussions on all results in Section 6.2 - and not two as the reviewer has mentioned in the review.
>
> Furthermore, although Waterbirds only have 2 classes, it is a dataset specifically designed to study spurious correlations [1, 3, 4], long-tail distributions, etc., as the dataset is created in such a way that the train is balanced while the test is extremely skewed.
>
> Additionally:
>
> - **GeoDE**: A dataset with 40 classes and additional geo-location labels for each sample (see paper lines 350-353).
> - **Aircrafts**: A dataset with 100 classes with high intra-class similarities, previously shown to be challenging for transformer model finetuning.

---

> > ### Author Response · Authors · 2024-11-25
> > **contd...**
> >
> > ### Experimental Setup
> >
> > To further elaborate on the scale of our experimental setup:
> >
> > - We use **3 datasets** and **3 models**:
> >   - ViT (pretraining with supervised learning)
> >   - CLiP (pretraining with contrastive vision and language learning)
> >   - DiNO (pretraining with SSL objective)
> > - For each model, we:
> >   - Try **5 different LoRA ranks**, with and without the fairness regularizer.
> >   - Perform full-finetuning experiments, with and without the regularizer.
> >   - Use **3 seeds** for each experiment.
> >
> > This results in ~200 runs, where each run takes **2-6 hours on GPU clusters** we have access to.
> >
> > Additionally:
> >
> > - These ~200 runs are based on **best hyperparameters**, searched separately for each rank, regularizer setting, model, and dataset across different regularizer coefficients.
> > - This totals to **more than 6000 experiments**, each with an average run time of 3 hours on GPU clusters with A100s.
> >
> > Hence, we believe that, contrary to the reviewer’s comment, our experiments were comprehensive and involved significant computational resources.
> >
> > ### Reporting Majority Group Performance
> >
> > We understand the reviewer’s concern regarding us not reporting the performance of the majority group. This decision was due to two main reasons:
> >
> > 1. For fairness evaluations, we are generally more concerned about what happens to the minority (or long-tails) than the majority [1, 3, 5, 7].
> > 2. With the min, avg, and overall performance metrics, the majority group performance can be inferred. If the majority group suffered, the average performance would have reduced, given the skewed dataset distribution.
> >
> > Examples for convenience:
> > - In the Waterbirds dataset:
> >   - Fair FFT: 89.98%
> >   - FairLoRA: 93.32%
> > - In GeoDE:
> >   - Fair FFT: 99.64%
> >   - FairLoRA: 99.76%
> >
> > As clearly seen from these examples, the best group is not degraded; in fact, it performs better with FairLoRA.
> >
> > If the reviewer still thinks it would be valuable to report these measurements, we are open to adding new tables in the appendix with all the measurements, including the max group.
> >
> > ---
> >
> > ## Novelty
> >
> > We thank the reviewer for pointing out that our solution is simple yet effective. The motivation for this formulation and why the degenerate solution wouldn’t occur is outlined in section 4.1, particularly lines 195-199. If the reviewer finds that explanation insufficient, we can further elaborate on it.
> >
> > More importantly, the novelty of our paper extends beyond merely modifying the loss function with fairness regularization during fine-tuning.
> >
> > ### Key Contributions:
> >
> > - We are the first to introduce **fairness in LoRA for vision tasks**.
> > - We offer a comprehensive evaluation of the impact on various fairness metrics by examining different ranks, data distributions, and architectures.
> >
> > ---
> >
> > ## Clarity of the Paper and the Story
> >
> > We have updated the following:
> >
> > - **Definitions of group** in Section 5 (lines 224-225, bolded text).
> > - Added equation numbers.
> > - Expanded discussion on previous work in Sections 7.1 and 7.2 (bolded text in lines 503-504 and 515-516).
> >
> > Thank you for pointing out the small font size in the figures. We will fix it in the camera-ready version of the paper.
> >
> > ---
> >
> > ## Conclusion
> >
> > We would like to thank the reviewer for their comments. We have addressed and clarified all the queries posed by the reviewer, particularly about the breadth and depth of our experiments, dataset choices, contributions, and novelty.
> >
> > We kindly request the reviewer to reconsider the current score. We would be happy to elaborate further on any clarification points if needed.
> >
> > ---

---

> > > ### Author Response · Authors · 2024-11-25
> > > **(Contd...) References**
> > >
> > > ## References
> > >
> > > [1] Pezeshki, M., Bouchacourt, D., Ibrahim, M., Ballas, N., Vincent, P., and Lopez-Paz, D., 2023. Discovering environments with XRM. arXiv preprint arXiv:2309.16748.
> > > [2] Sagawa, S., Koh, P.W., Hashimoto, T.B., and Liang, P., 2019. Distributionally robust neural networks for group shifts: On the importance of regularization for worst-case generalization. arXiv preprint arXiv:1911.08731.
> > > [3] Deng, Y., Yang, Y., Mirzasoleiman, B., and Gu, Q., 2024. Robust learning with progressive data expansion against spurious correlation. Advances in Neural Information Processing Systems, 36.
> > > [4] Ming, Y., Yin, H., and Li, Y., 2022, June. On the impact of spurious correlation for out-of-distribution detection. In *Proceedings of the AAAI Conference on Artificial Intelligence* (Vol. 36, No. 9, pp. 10051-10059).
> > > [5] Wang, S., Narasimhan, H., Zhou, Y., Hooker, S., Lukasik, M., and Menon, A.K.,
> > > [6] Hashemizadeh, M., Ramirez, J., Sukumaran, R., Farnadi, G., Lacoste-Julien, S. and Gallego-Posada, J., 2023. Balancing act: Constraining disparate impact in sparse models. arXiv preprint arXiv:2310.20673.
> > > [7] Hardt, M., Price, E. and Srebro, N., 2016. Equality of opportunity in supervised learning. Advances in neural information processing systems, 29.
> > > [8] Nwatu, J., Ignat, O. and Mihalcea, R., 2023. Bridging the digital divide: Performance variation across socio-economic factors in vision-language models. arXiv preprint arXiv:2311.05746.
> > > [9] Richards, M., Kirichenko, P., Bouchacourt, D. and Ibrahim, M., 2023. Does Progress On Object Recognition Benchmarks Improve Real-World Generalization?. arXiv preprint arXiv:2307.13136.

---

> ### Author Response · Authors · 2024-12-02
> **Summary of the above response and request for engagement**
>
> Dear Reviewer,
>
> We sincerely thank you for your thoughtful feedback and look forward to engaging with you during the remaining discussion period. We have thoroughly addressed all your concerns to the best of our ability and would greatly appreciate your continued engagement during the final discussion phase.
>
> In particular, we have clarified the impact of low variance in loss as an optimization problem, the definition of fairness, and the thoroughness of our experiments across models and datasets in our response: [https://openreview.net/forum?id=pB3KeBCnQs&noteId=xMWKVreJFz](https://openreview.net/forum?id=pB3KeBCnQs&noteId=xMWKVreJFz). Additionally, we have elaborated on the experiments, reported the best group performance as requested, improved the paper’s clarity, and emphasized the novelty of our contributions in this response: [https://openreview.net/forum?id=pB3KeBCnQs&noteId=q5azYNRZ7z](https://openreview.net/forum?id=pB3KeBCnQs&noteId=q5azYNRZ7z). We believe these updates directly address the raised concerns.
>
> We hope these clarifications and improvements are helpful and will be reflected in your evaluation. Thank you once again for your time and thoughtful review!

---

### Author Response · Authors · 2024-12-02
**Combined response to all reviewers highlighting all the changes**

We sincerely thank the reviewers for their constructive feedback. Below, we address the main comments, grouped by themes, and provide clarifications (on top of the individual more detailed responses).

## 1. Low Variance of Loss and Fairness

We agree that low variance in loss does not inherently ensure fairness and could lead to a degenerate solution. To address the issue of degenerate solution, **Equation (1)** in Section 4.1 combines an ERM term with the fairness regularizer based on variance and to ensure fairness the reducing variance in combination with the usual ERM objective tends to improve the worst group performance without harming the overall performance.

This ensures that the model simultaneously optimizes for accuracy and group fairness.

---

## 2. Fairness Definitions and Metrics

We employ multiple fairness metrics to provide a holistic evaluation:

- **Min F1 per class** and **min recall per class**: Evaluate fairness across classes.
- **Equal Opportunity Difference (EOD)**: Measures fairness across sensitive attributes.
- **Sensitive attribute prediction**: Tests fairness across tasks, such as spurious correlations in the Waterbirds dataset.

For example, on the Waterbirds dataset, our FairLoRA model achieves:
- **ΔF1**: \(18.58 +/- 1.15\).
- **Maximum EOD**: \(37.84  +/- 0.43\) (see Table 2).

---

## 3. Datasets and Domains

The datasets used cover diverse challenges:

- **GeoDE**: A long-tailed dataset with 40 classes and geo-location metadata.
- **Waterbirds**: A spurious-correlation dataset with synthetic sensitive attributes.
- **Aircrafts**: A dataset with high intra-class similarity.

We have clarified group and sensitive attribute definitions in Section 5.

---

## 4. Experimental Setup and Scale

Our experiments span:
- **3 datasets**, **3 architectures** (ViT, CLiP, DiNO), and varying ranks with/without fairness regularization.
- ~200 runs over 3 seeds, supported by hyperparameter searches involving **6,000 experiments**, consuming ~3 GPU hours per experiment.

This further demonstrates the thoroughness of our evaluation as highlighted by the majority of reviewers and addresses concerns about insufficient experiments raised by reviewer 42sa.

---

## 5. Novelty and Contribution

Our contributions include:
1. Introducing fairness into LoRA for vision tasks.
2. Providing insights into fairness-specific challenges, such as non-monotonic behavior with varying LoRA ranks (see Figure 5).
3. Highlighting the importance of evaluating fairness across multiple metrics
4. Exploring the impact of pre-training models, pre-training data and pre-training techniques when used with FairLoRA

---

## 6. Majority Group Performance

Fairness studies often focus on minority groups. However, majority group performance can be inferred from overall metrics. For example, on the Waterbirds dataset:

- FairLoRA improves the best group accuracy from **89.98%** to **93.32%** without harming the fairness objective.

We are happy to add explicit majority group metrics in the appendix, if requested.

---

## 7. Visualization, Clarity and Related Work

We have added examples of visualization of the impact of FairLoRA on the distribution shift as requested by the reviewer, details can be found in - https://openreview.net/forum?id=pB3KeBCnQs&noteId=F5C79RolEe

We have clarified group definitions in Section 5 and expanded discussions on related work in Sections 7.1 and 7.2. Minor visual issues, such as font sizes in figures, will be fixed in the final version.

---

## 8. Future Directions

We thank the reviewers for their suggestions on applying FairLoRA to large language models and segmentation tasks. While beyond the scope of this work (agreed by the reviewer as well), these are excellent directions for future exploration (added to Section 8).

---

## Conclusion

We have addressed all reviewer comments and revised the manuscript accordingly. We strongly believe that the feedback has improved the quality and readability of our paper and thank the reviewers for their invaluable contribution. On that note, we kindly request the reviewers to reconsider their current score evaluations. We are happy to provide further clarifications if needed during the reminder of the discussion period and also make adjustments in the camera ready version.

---

### Meta-Review · Area_Chair_WnxH · 2024-12-17

**Metareview:**

The paper received mixed reviews from the reviewers. While the authors actively provided explanations and clarifications to the manuscript, some issues still remain (e.g., insufficient experiments on visualization and datasets). Overall, the ratings remain marginally below the threshold for acceptance, and thus the paper is not accepted for publication.

**Additional Comments On Reviewer Discussion:**

I think the authors did a good job during rebuttal, but not all the reviewers agreed to raise their ratings.

---

### Decision · Program_Chairs · 2025-01-22

Reject